# Does decision confidence reflect effort?

**Nobuhiro Hagura**[1,2]*, **Jamal Esmaily**[3,4], **Bahador Bahrami**[3,5,6]

**1** Center for Information and Neural Networks, National Institute of Information and Communications Technology, Osaka, Japan, **2** Graduate School of Frontiers Biosciences, Osaka University, Osaka, Japan, **3** Department of General Psychology and Education, Ludwig Maximilians University Munich, Munich, Germany, **4** Graduate School of Systemic Neurosciences, Ludwig Maximilians University Munich, Munich, Germany, **5** Centre for Adaptive Rationality, Max Planck Institute for Human Development, Berlin, Germany, **6** Department of Psychology, Royal Holloway University of London, Egham, Surrey, London, United Kingdom

* n.hagura@nict.go.jp

## Abstract

Goal directed behaviour requires transformation of sensory input to decision, and then to output action. How the sensory input is accumulated to form the decision has been extensively studied, however, the influence of output action on decision making has been largely dismissed. Although the recent emerging view postulates the reciprocal interaction between action and decision, still little is known about how the parameters of an action modulates the decision. In this study, we focused on the physical effort which necessarily entails with action. We tested if the physical effort during the deliberation period of the perceptual decision, not the effort required after deciding a particular option, can impact on the process of forming the decision. Here, we set up an experimental situation where investing effort is necessary for the initiation of the task, but importantly, is orthogonal to success in task performance. The study was pre-registered to test the hypothesis that the increased effort will decrease the metacognitive accuracy of decision, without affecting the decision accuracy. Participants judged the direction of a random-dot motion stimuli, while holding and maintaining the position of a robotic manipulandum with their right hand. In the key experimental condition, the manipulandum produced force to move away from its position, requiring the participants to resist the force while accumulating the sensory evidence for the decision. The decision was reported by a key-press using the left-hand. We found no evidence that such incidental (i.e., non-instrumental) effort may influence the subsequent decision process and most importantly decision confidence. The possible reason for this result and the future direction of the research are discussed.

## Introduction

Traditionally, perceptual decision making and motor control have been studied as separate domains. This tradition has benefited researchers by simplifying assumptions that regard action as a mere output channel for reporting the higher-order decisions. However, recent developments at the theoretical and empirical level have started to demonstrate that parameters of action, which are not directly related to the sensory input per se, can influence the

**Data Availability Statement:** Content of the pre-registration is available at https://osf.io/tykcg. The associated data is accessible at https://osf.io/be4u5/.

**Funding:** NH is supported by Japan Society for the Promotion of Science (Kakenhi: 20H00107,

21H00314) and Japan Science and Technology Agency (ERATO: JPMJER1801). BB and JE are supported by the European Research Council (ERC) under the European Union's Horizon 2020 research and innovation programme (819040 - acronym: rid-O). BB is supported by the NOMIS foundation. The funders had no role in study design, data collection and analysis, decision to publish, or preparation of the manuscript.

**Competing interests:** The authors have declared that no competing interests exist.

perceptual decision itself [1]. For example, two studies [2, 3] showed that when the response for one of the two options in a perceptual decision is made physically more effortful, participants tend to avoid the decision requiring more effort, without any change in their sensitivity at the task. Another study [4] showed that participants became more confident when a perceptual decision required more physical effort to express.

Effort can be classified as instrumental to a given task when it is necessary for better performing that task. Conversely, effort may be needed to perform the task and yet be irrelevant to the chances of success. This latter type of effort may be classified as incidental effort [5]. For example, running faster is instrumental to winning a race but thinking hard about a difficult math problem while running the same race is incidental to the race outcome. Thus, instrumental effort is directly linked to the improvement of the decision, such as in the situation where the amount of effort is proportional to the amount of evidence one can acquire. In contrast, incidental effort does not contribute to the quality of the decision per se.

A key common feature of the studies mentioned above [1–5] is that they all employed incidental effort. The amount of effort expenditure did not benefit for improving the decision accuracy. Thus, their counterintuitive observation was to show that the effort influenced the decision process even though it was irrelevant to the task, thus indicating the tight interdependency of action and perceptual decisions.

In those studies, examining the influence of incidental physical effort on the particular decision the participant has (presumably already) made meant that physical effort was required *after* the decision. However, the post-decision period is not the only period in which the decision and the action can interact [6]. In real life, as well as under laboratory conditions, there are situations where a decision is required *while* performing an effortful action. Imagine a monkey hanging from a branch with one hand and using the other hand to pick a fruit deliberating which fruit to pick. Here, the effort (i.e. keeping herself from the branch) is required *during* the deliberation period of the decision-making, during which perceptual evidence is being accumulated to reach the decision. Importantly, the effort is necessary to stay in the task, but does not directly change the quality of the decision evidence, thus, is incidental. The effect of such physical effort exerted during the pre-decision period on decision making is not well understood. In this study, we examined whether the physical effort required during the deliberation period of the decision making influenced the subsequent decision and the confidence for the decision.

Several different lines of research on the role effort can be employed to formulate contrasting hypotheses for the expected results. In a recent review, Inzlicht, Shenhav and Olivola [7] have done a wonderful job of describing what they called the effort paradox. On the one hand, prominent models in economics and cognitive sciences hold that effort is aversive. Animals (including humans) prefer to do as little as possible for a given reward. On the other hand, numerous studies spanning all the way from pigeons and rodents to humans, have shown that all of these various animals value effort, in and of itself. People go mountain climbing, run 7 marathons in 1 week, and spend hundreds of hours learning a musical instrument as a hobby. While few would contribute to the cost of a ticket for you to join a charity cocktail party, it is much easier to raise money for joining a charity triathlon. To the standard models of economics that seek to minimise effort this does not make sense: people who love you should be happier if you enjoyed yourself in the cocktail party. It is therefore not surprising that several contradictory hypotheses can be formulated when considering the impact of effort on decision making.

The common currency [8–13] hypothesis proposes we try to maintain the balance between expected accuracy and cost of evidence accumulation. This cost consists of time and effort. In this view, optimal decisions are the ones that achieve a reasonable level of accuracy at an acceptable cost. Increasing the cost, then, should reduce the bar for what is the acceptable level

of accuracy. This would predict that under higher effort, participants would adjust evidence accumulation to collect less evidence, leading to lower accuracy, faster reaction times and lower confidence under high effort. Here, the hypothesis assumes that the brain treats the duration of effort investment during the deliberation as instrumental, and sacrifices accuracy over effort.

The second hypothesis is derived from the ideas of 17th century Dutch philosopher Baruch (Benedictus) Spinoza. In his book of Ethics [14] Spinoza proposed that individuals are motivated to be the adequate cause of their states. A consequence of this kind of motivation is that the power of actions and the satisfaction driven from the choices that those actions execute are inextricably intertwined. Spinoza's idea of the association between power and joy is reflected brilliantly in the social psychological research on the role of self-determination and control in self efficacy [15–17]. Recent findings in neuroeconomics have shown that having a choice, in and of itself, is rewarding [18, 19]. This account is perhaps most directly supported by the evidence from a recent study by Turner et al [4] who showed that participants expressed higher confidence when deciding required more physical effort. In this view, exerting effort would have a metacognitive, evaluative impact on the decision process that does not affect the accuracy and reaction time but the *evaluation* of the decision. This hypothesis proposes that evidence accumulated under high effort would be evaluated as more valuable or of better quality because it has cost the participants more and the participants has obtained the evidence through effort and control. Thus, higher effort would impact neither accuracy nor reaction time per se but increase decision confidence. In this hypothesis, the brain ignores the direct benefit of the effort to the task, thus treating it as incidental effort, but using the invested effort as evidence for postdictive evaluation of the task performance.

Finally, an interference hypothesis is also plausible and would predict that higher effort would specifically interfere with the relationship between confidence and accuracy. Fleming and colleagues [20] examined the impact of transcranial magnetic stimulation in interfering with the motor execution of perceptual decisions. They found that, within a narrow and carefully designed range of parameters, one can find a TMS stimulation protocol that leaves perceptual decision accuracy, reaction times and average confidence intact but, critically, disturb metacognitive sensitivity, which is the delicate balance between reported confidence and probability of correct responses. The interference hypothesis driven from these findings, would predict lower metacognitive sensitivity under high (vs low) effort. Apart from using effort (rather than TMS), an important difference between our experimental design and those of Fleming and colleagues is that our manipulation takes place *during the perceptual* evidence accumulation stage whereas in their work, TMS was applied after the perceptual information had been delivered to the participants. Here, the prediction is that the incidental effort distracts the evaluation of one's task performance.

To test these hypotheses, our participants judged the direction (leftward or rightward) of a dynamic random dot motion stimulus [21, 22] whose difficulty (i.e. coherence) level varied randomly across trials. The decision and the subjective confidence of the decision was reported by a key-press using their left-hand. While viewing the dots motion, participants held a handle with their right-hand. The handle produced force to move-away from its initial position, and the participants had to maintain its position by resisting the force (Fig 1). We designed this setup to examine how an ongoing isometric physical effort may interact with the deliberation process of decision making, independent of the decision itself and possible outcomes.

Since we did not have any hypothesis regarding the direction of the effect, a pilot experiment was performed using a models sample of participants. Based on the exploratory analysis performed on the pilot data, the present study is pre-registered to test the following specific predictions (https://osf.io/tykcg). First, we predict that the metacognitive accuracy [23], the measure of how accurately the confidence is related to the decision accuracy, will decrease

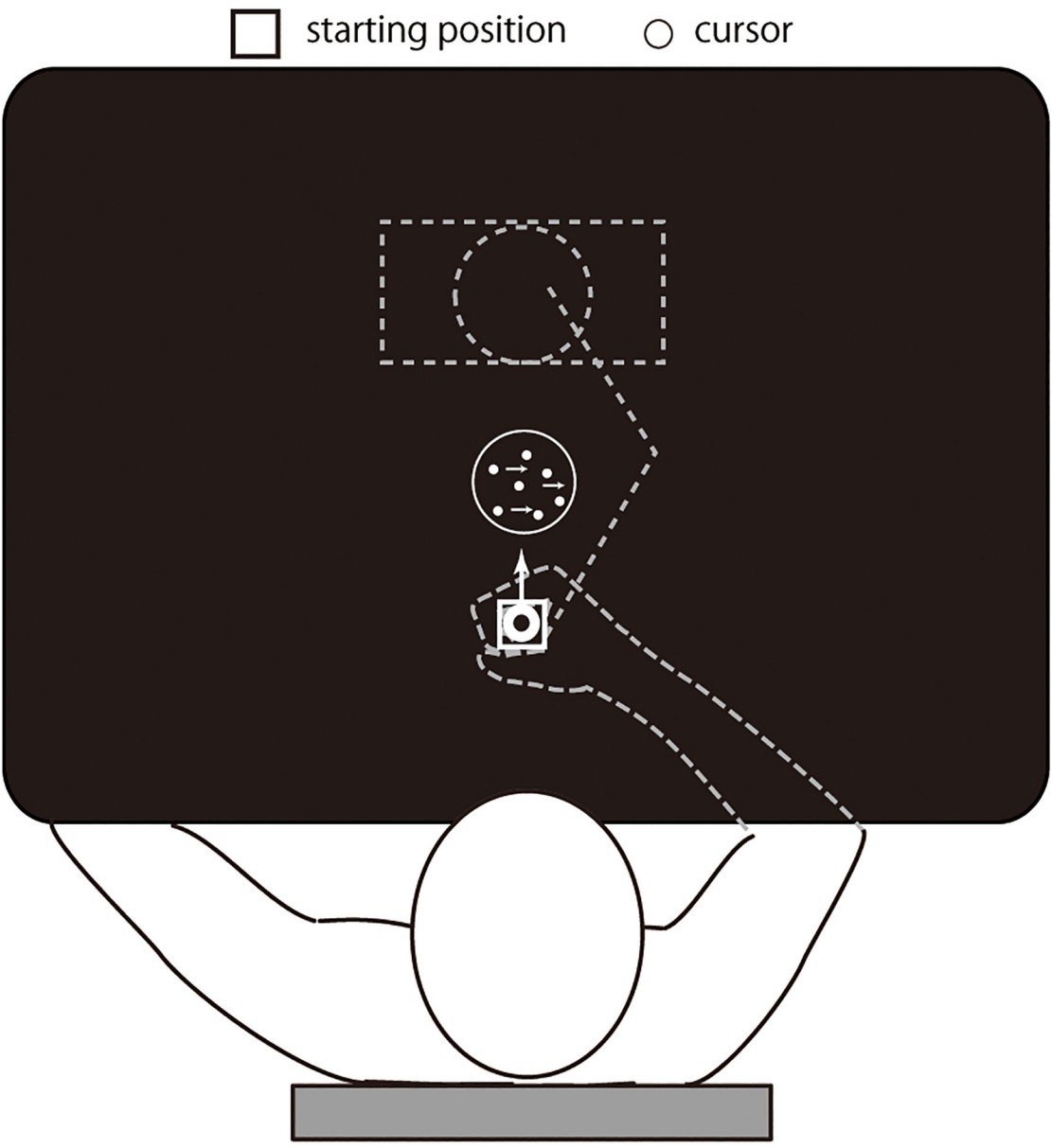

**Fig 1. Schematic drawing of the experimental setup.** Participants held a robotic handle underneath the screen. In each trial, a random-dot motion stimulus was presented in the centre of the screen. Throughout the trial, they were required to maintain the handle position, indicated as a cursor position on the screen, in the starting position. In the force-condition trials, a force was applied to the handle towards the direction away from the participant (i.e. the arrow in the drawing).

when the required effort increases. Second, we predict that the physical effort during the deliberation period will not modulate the reaction time nor the decision accuracy of the task.

## Method

### Participants

A total of 20 volunteers (12 females, mean age: 23.7, s.d.: 4.2 (21–39)) participated in the study. Sample size was determined using the result of the pilot experiment (see below). All were

right-handed with normal or corrected-to-normal vision, with no prior history of psychiatric problems or medication. Experiments were undertaken with the understanding and written consent of each participant in accordance with the Code of Ethics of the World Medical Association (Declaration of Helsinki), and with approval of the National Institute of Communications Technology (NICT) ethical committee (B210042202). Participants were recruited through local advertisements used in the Centre of Neural Networks (CiNet), and were paid 3000 JPY (approximately 10 Euros per hour) for participation. No adverse events occurred during the experiment.

## Estimation of the sample size

Data of 14 participants from a pilot experiment was analysed for the sample size estimation (see below for the detail of the pilot experiment). The aim of the pilot experiment is to perform an exploratory analysis to generate a hypothesis about the direction of the effect, as well as estimating the sample size to verify the effect in the main experiment.

In this experiment, the metacognitive accuracy for the random-dot motion decision became lower when maintaining the handle position against the force compared to when there was no force applied to the handle (effect size = 0.76). Based on this result, we estimated the required sample size using the power analysis (G-power), targeting for alpha = 0.5 and power = 0.8. Taking into account the estimation noise, we pre-set the sample size to 20.

## General setup

Participants were seated comfortably in front of a screen placed horizontally in front of them, which prevented the direct vision of their hands (Fig 1). The visual stimulus was presented on the screen using a projector placed above the screen. The viewing distance was 50cm. To maintain the viewing distance, the upper trunk was constrained by a harness attached to the chair.

During the experiment, participants were asked to hold the handle of the manipulandum with their right-hand (PHANToM Premium 1.5 HF, SensAble Technologies, Woburn MA, USA). The handle position was displayed as a white cursor (a circle, 6 mm in diameter) on a black background on a horizontal screen located above their hand. The movement of the handle was constrained to a virtual horizontal plane (10 cm below the screen) that was implemented by a simulated spring (1.0 kN/m) and dumper (0.1 N/ms-1).

In the centre of the screen, a random-dot motion stimulus was presented [22] (Fig 1). In a 7 deg diameter circular aperture, dots were presented at a density of 3.5 dot/deg2. The speed of the dots was 10 deg/s. For each trial, either 3.2%, 6.4%, 12.8%, 25.6%, or 51.2% of the dots moved coherently to the left or the right. All other dots moved in a random direction, picked for each dot separately between 0 and 360 deg.

Both the visual stimulus and the robotic manipulandum were controlled by an in-house software program developed using C++.

## Study design and the procedure

The study was conducted as 5 (sensory evidence level) x 2 (physical effort level) within-subject factorial design. For the factor of sensory evidence, random-dot motion with 5 different levels of motion coherence (3.2% 6.4% 12.8% 25.6% 51.2%) was prepared. The direction of motion was either leftward or rightward. For the factor of physical effort level, either the handle was pushed away from the body with the force of 6N (force condition) or the force to the handle was absent (no-force condition).

To start each trial, the participant held a handle of a robotic manipulandum with their right hand in the starting position. The trials started 500 msec after setting the handle position. In

the force condition trials, a force was applied to the robotic handle towards the direction moving away from the participant. Participants were asked to maintain the handle in the starting position throughout the task, meaning that the participant had to apply an equal force (6N) in the opposite direction. Therefore, if the participants did not produce 6N toward the direction of one's body to resist the manipulandum movement, the participants' hand would be passively moved towards the direction of the force from the starting position, or the manipulandum would slip away from the participants' hand (and collide to the wall). In the no-force condition trials, the force was absent and the participant could effortlessly hold the handle in the starting position. Thus, in this setup, the amount of positional deviation of the manipulandum from the starting position would directly indicate the failure to produce the required amount of force.

In both trial types, once the participant had maintained the handle position for 2000 msec after the start of the trial, then a random-dot motion stimulus was displayed on the screen. Motion coherence varied from trial to trial. Participants were asked to decide about the net direction of the dot-motion and indicate their decision, as fast and accurately as possible by pressing one of the two keys on the keyboard with their left hand. Visual stimulus and the force were removed immediately after the decision (key-press), or if the decision was not made within 2000ms. Then, participants reported the confidence level of the decision on a scale of 0–6. Reaction time, decision accuracy and confidence were our key dependent variables. Kinematic data of the handle position during the task were also measured and recorded. Each participant went through 24 trials of each combination of the condition (i.e., 2 motion directions x 5 coherence x 2 effort levels). One block consisted of 60 trials and there were 8 blocks. The order of the trials was pseudo-randomized within each block of each participant.

## Pilot experiment

An earlier version of the experiment was treated as a pilot experiment, in order to estimate the sample size (see above). In this pilot experiment, 15 participants (5 females, mean age = 22.4) volunteered, but 1 was excluded from the analysis, due to the high proportion of non-responding trials. Three different force levels (0N, 2.5N, and 5N) were set, with 200 trials for each force level. The results of this pilot experiment are available in S1 Fig in S1 File.

## Data analysis

Prior to the parametric analysis, we confirmed that the data in each condition did not violate the normality assumption using the Kolmogorov-Smirnov test (p>0.05 for all cases). For the main analysis, Generalised linear mixture model (GLMM) was applied to the reaction time, decision accuracy and the confidence data. For all of the GLMM models applied here, each participant was considered as a random intercept. Metacognitive accuracy was calculated as in the previous literature [23]. In short, metacognitive accuracy is quantified by the interrelationship of the confidence rating and the accuracy of visual judgments using the type II receiver operating characteristic (ROC) curve, which characterized the probability of being correct for a given level of confidence [20, 24, 25]. To construct the ROC curve, p (confidence = i | correct) and p (confidence = i | incorrect) were calculated for all i, and were then transformed into cumulative probabilities and plotted against each other. The area underlying the ROC curve quantified the metacognitive accuracy. Metacognitive accuracy was compared between different force conditions using paired t-test. We also calculated the metacognitive accuracy for each low (3.2 and 6.4%) and high (12.8, 25.6 and 51.2%) coherent motions, to check if there is any difference in the effort linked modulation pattern for different uncertainty levels.

Inclusion and exclusion criteria as well as policy for dealing with missing data were pre-registered at OSF https://osf.io/tykcg. In detail, any trial with the decision taking more than 2 seconds, and any trial with the handle movement exceeding over 5cm was excluded from the analysis. Any participant's data with the proportion of the excluded trials exceeding 15% of the whole trial was discarded from the analysis.

**Prediction.** In this study, we test our pre-registered hypothesis. First, we predict that the Metacognitive accuracy will be affected by the physical effort delivered during the deliberation period of the decision-making. Metacognitive accuracy will be reduced by physical effort, compared to when no physical effort is required. Second, we predict that the preceding effort will not influence the objective decision accuracy, or the time it takes for the decision (reaction time).

## Results

### Data exclusion

On average, 3.4% of the trials in the non-force condition and 6.7% of the trials in the force condition were excluded from the analysis. None of the participants exceeded the pre-defined exclusion criterion, therefore, were not excluded from the analysis.

### Kinematic data

Since participants were asked to maintain the position of the handle during the task, large deviation of handle position from the starting position would indicate the failure to produce the required amount of force. By analyzing the kinematic data, we confirmed that participants were indeed producing force in the force condition compared to the non-force condition.

In 98.5% of all of the trials, deviation was in the range within 50mm, which was within the pre-defined inclusion criterion (i.e. 1.5% of the data was excluded from the further analysis for this reason). For the rest of the trials, the amount of deviation was at the negligible level. During the force trials, the deviation was on average 17.5mm (s.d. 10.6mm) towards the direction of the force, and 8.9mm (s.d. 6.5mm) towards the right. For non-force trials, it was -2.0 mm and 1.3 mm, respectively.

We also calculated the within-trial fluctuation of the handle position (i.e. variance of the handle movement), which should be proportionally scaled by the amount of force output due to the signal dependent noise [26]. Indeed, the variance was significantly higher for the force condition compared to the non-force condition (paired-t test; t(19) = 2.56, $p$ = 0.019). Taken together, these data indicate that participants were indeed producing a significant amount of force during the force condition compared to the non-force condition (S2 Fig in S1 File).

### Decision accuracy, decision confidence and the reaction time

Decision accuracy (Fig 2a), confidence data (Fig 2b) and reaction times (Fig 2c) were compared between two different force conditions under different uncertainty levels. First, all three variables were modulated by the difficulty of decision (i.e. motion coherence levels) (p<0.001, GLMM, see S1 Table in S1 File for details of the statistical values). As expected, this shows that the participant's response is manipulated by the different coherence level (i.e. difficulty level) of the visual stimulus.

Second, decision accuracy and the confidence ratings did not differ between the two force conditions ($p$> 0.1, GLMM (S1 Table in S1 File)), which followed our pre-registered prediction. However, the reaction time became slightly faster for the force condition compared to the non-force condition (average RT; 1028ms for the force condition, 1047 for the non-force

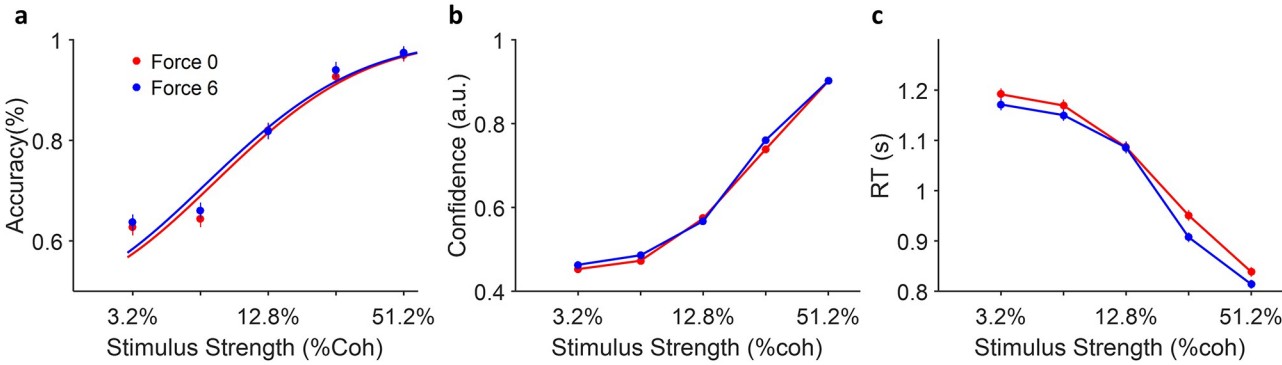

**Fig 2. Comparison of behavioural data in different force conditions. a:** Accuracy of participants did not change by force conditions (red = force 0N and blue = force 6N). The curve is the weibull cumulative distribution fitted on the averaged data points of each condition (dots). **b:** Same as **a** but for confidence. **c:** Same as **a** but for RT. The difference here is significant. Error bars indicate the standard error of means (s.e.m) across all trials.

condition, $p<0.001$, GLMM (S1 Table in S1 File)). Note that, since faster reaction time during the force trials did not change the accuracy. This implies that our participants did not use the effort instrumentally, for example to prolong stimulus presentation actively in order to sample more evidence before responding.

## Metacognitive accuracy

The metacognitive accuracy [20], i.e. how the confidence is correctly associated with the accuracy of decision, was calculated and compared between different force levels. We found no significant difference in metacognitive accuracy between different conditions ($t(19) = 0.22$, $p = 0.83$; Fig 3). Thus, this result did not support our pre-registered prediction. Furthermore, we also confirmed that such non-significant result holds even when the metacognitive accuracy was calculated independently for high and low motion coherence levels (S3 Fig in S1 File).

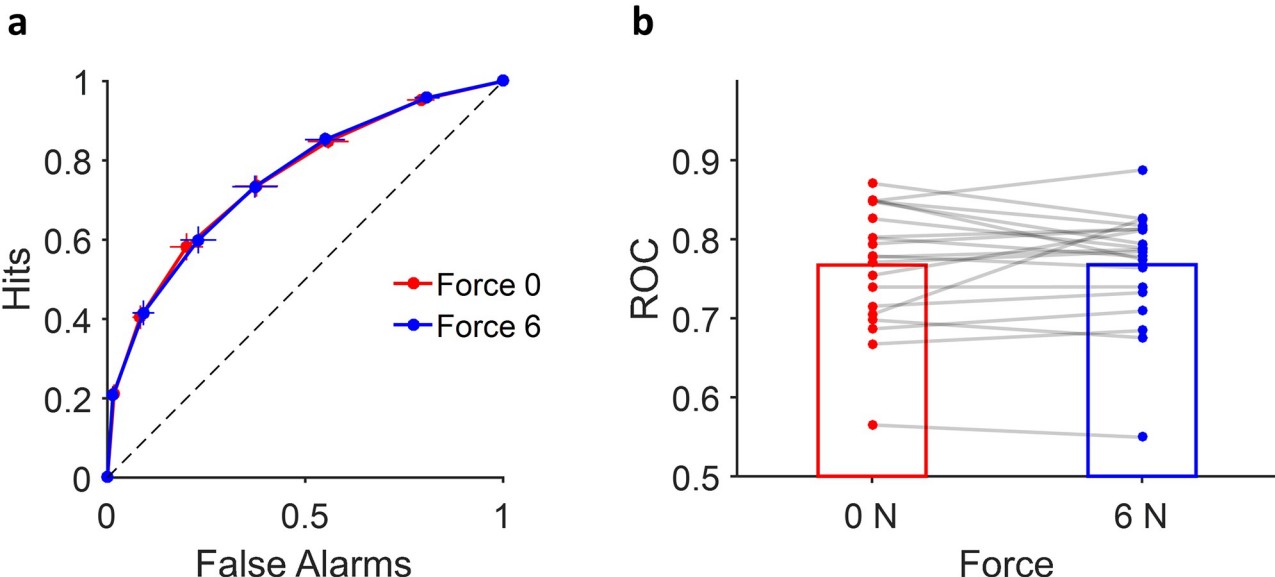

**Fig 3. Metacognitive accuracy based on force conditions. a:** ROC curve of subjects in 0N (red) and 6N (blue) conditions. **b:** The area under the curve of ROC. Each dot and line represent different participant. Error bars indicate the standard error of means (s.e.m) across participants for both the hit rates and false alarm rates.

## Discussion

In this study, we tested if exerting physical effort during the deliberation period of the perceptual decision-making influences the subsequent decision process. We proposed three different hypotheses that made contrasting predictions about this question. Based on a pilot experiment, in the pre-registration we predicted that the physical effort will reduce metacognitive accuracy, without affecting decision accuracy, reaction time or confidence. However, the hypothesis was not supported by the data. In our experimental setup, we found no evidence that the physical effort during the deliberation period could influence the subsequent metacognitive process. We report this negative result together with the pre-registered methods to help avoid file-drawer problems.

One reason for the discrepancy between the pilot and the main study results may be the few differences in the design elements that we changed in the main study (see S1 File). In the pilot experiment, we examined three force levels (0N, 2.5N, and 5N), whereas in the main experiment, there were two force levels (0N and 6N). Accordingly, the number of trials per condition also differed between two experiments. These choices were made to increase the power and sensitivity of the experimental paradigm and there is no principled reason to believe that the choices in the main study were problematic. In any case, if such trivial differences were indeed the cause of the failure to replicate the pilot experiment, this would indicate that the effect of physical effort on metacognition is not robust, at least in the given setup of the experiment.

As with the nature of any negative result, our findings do not necessarily discredit the hypothesis. Physical effort during the deliberation period may or may not influence the perceptual decision in general. We examine several issues worth considering for future studies. First, the pragmatic choices made for experimental design may be responsible for the negative results. The amount of force could have been too low to induce a sufficient amount of effort to influence the participants' decision. In the current setup, we had to make sure that the amount of force is well within the range which the participants could muster and maintain holding the handle. This was because the slip of the handle from the participants' hand could cause the handle to crash into the surrounding devices. This restriction of force level may have made the result unstable. Stronger force level may be necessary to overcome this problem, if using passive resistance to the force as a way to manipulate the effort.

Another point to consider is that the force level we used did not affect the accuracy, indicating that the effort level remained non-instrumental to the task. Increasing the force may change this balance of instrumental vs. incidental in our task, which is an interesting question to pursue in the future studies.

Another pragmatic issue was the way the force was produced in the experiment may have made a difference. We chose passive resistance to the force to manipulate the physical effort, because this allowed us to precisely control the required force level. However, an alternative way would have been to make the participants perform an approach movement, actively producing the force to a set level. One may argue that active effort may impact information seeking more directly, thus, making the effort more instrumental. Moreover, there could have been an effector dependency of effort on decision making. Previous studies have manipulated the effort on effector directly involved in the decision, such as by adding physical perturbation to that effector [2, 3] or by electrically/chemically perturbing the neural substrate controlling that effector (eye-movement) [6, 27]. However, in our study, the effector that resisted the force (right-hand) and the effector expressing the decision by the button press (left-hand) differed. Even though the force was terminated after the decision, the relationship between the decision and the force was indirect. Such effector dependency may have played a role in our experiment.

**Table 1. An overview of effort hypotheses based on different behavioural features.** In Table 1 ↓ shows decrease of the behavioural feature with respect to increasing the effort, ↑ shows increase and × shows no change;—indicates no prediction.

| Hypothesis/Behavioural prediction | Accuracy | Reaction time | Confidence | Metacognitive precision |
|---|---|---|---|---|
| Hypothesis 1 | ↓ | ↓ | ↓ | - |
| Hypothesis 2 | × | × | ↑ | - |
| Hypothesis 3 | × | × | × | ↓ |
| Our Data (Pilot) | × | × | × | ↓ |
| Our Data (Replication) | × | ↓ | × | × |

Having discussed the potential reasons for interpreting our results as false negatives, it is also important to discuss the plausible interpretation of them as true negatives too. The three main hypotheses that we raised in the introduction have been summarized in Table 1 together with their predictions for the 4 outcome measures of accuracy, reaction time, confidence and metacognitive precision. Our pilot data most closely favored hypothesis 3 [20] i.e., that the effort influences the metacognitive precision without affecting the perceptual decision itself. Indeed, in our main experiment, physical force did not change the parameters of perceptual decisions *per se*.

Surprisingly, we also found a significant decrease in RTs under the high effort condition without any change in the metacognitive sensitivity, choice accuracy or confidence. This isolated decrease in reaction time without any impact on accuracy and confidence points to an intriguing possibility. Under zero force condition, this finding implies, participants were not doing their optimal best but were, so to speak, complacent. When pushed by the force requirement, they stepped up and performed faster while at the same level of accuracy and confidence. Note that faster reaction times meant participants collected less perceptual evidence. This finding is not easy to square with previous works on speed-accuracy tradeoff (see [28–30]). More recent studies that have argued for a relationship between confidence and reaction time are similarly troubled by this finding. For example, Kiani and colleagues [31] showed that artificially prolonging RT, for example by inserting a delay between perceptual stimulus presentation and choice, decreases confidence. Taking their findings at face value, one would have predicted that in our experiment, faster reaction times under high force should have increased confidence, which is not borne by the data. Thus, we believe that our findings point to either a different mechanism not captured in our hypotheses, or more likely, a mixture of different mechanisms. An interesting avenue for future research is to explore how paradigms like ours could help find ways to persuade and motivate people to step up and do better.

Whether force could have acted as a motivating factor raises another important possible reason why our effort manipulation did not succeed as predicted. In a series of innovative experiments addressing the effort paradox (see Introduction), Clay and colleagues [32], gave their participants a challenging training task (i.e., a working memory test) requiring substantial cognitive effort. In the treatment condition, participants were rewarded proportionally to how much effort they put into the training task (and not by success or failure in doing the memory task). In the control condition, participants received a fixed reward. After this initial training, both groups completed a second, transfer task (math puzzles) for a fixed reward. The results showed that the participants in the treatment group were later more eager (vs. control) to exert mental effort: the treatment group voluntarily chose more difficult problems to solve. Effort-contingent rewards during training led to demand seeking in the transfer task. This result shows that in the transfer task, where effort is certainly not instrumental, the previous experience of effort-reward contingency can affect behavior. These findings raise interesting questions. For example, on a general level, are the findings specific to cognitive, mental effort or do

they generalise to physical effort too? With respect to our experiment, our participants received a fixed reward compensation that was independent of their performance. Therefore, our experiment may be considered similar to Clay and colleagues' control condition. Future experiments that examine the role of effort in decision making would do well to consider and perhaps include the role of reward explicitly in their experimental design, and to investigate the interaction between instrumental and incidental effort.

## Supporting information

**S1 File.**
(PDF)

## Acknowledgments

Authors are grateful to Drs. Masaya Hirashima and Kisho Ogasa for their help in setting up the experiment, and to Mari Koshimizu for helping the data collection.

## Author Contributions

**Conceptualization:** Nobuhiro Hagura, Bahador Bahrami.

**Data curation:** Nobuhiro Hagura, Jamal Esmaily.

**Formal analysis:** Nobuhiro Hagura, Jamal Esmaily.

**Funding acquisition:** Nobuhiro Hagura, Bahador Bahrami.

**Investigation:** Nobuhiro Hagura, Bahador Bahrami.

**Methodology:** Nobuhiro Hagura, Jamal Esmaily, Bahador Bahrami.

**Resources:** Nobuhiro Hagura.

**Software:** Nobuhiro Hagura, Jamal Esmaily.

**Visualization:** Jamal Esmaily.

**Writing – original draft:** Nobuhiro Hagura, Bahador Bahrami.

**Writing – review & editing:** Nobuhiro Hagura, Jamal Esmaily, Bahador Bahrami.

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
