## [Decision Letter · Decision Letter 0]

19 Jul 2022

PONE-D-22-12208Does decision confidence reflect effort?PLOS ONE

Dear Dr. Hagura,

Thank you for submitting your manuscript to PLOS ONE. After careful consideration, we feel that it has merit but does not fully meet PLOS ONE’s publication criteria as it currently stands. Therefore, we invite you to submit a revised version of the manuscript that addresses the points raised during the review process.

Please, revise the manuscript to clarify the validity of the experimental paradigm and to address the reviewers' concerns about  statistical analysis. Please, revise to disambiguate the roles of 

two different types of effort : (i) effort that is instrumental to achieving the current goal, versus (ii) effort that is incidental to the task and their associated predictions/interpretations. 

We look forward to receiving your revised manuscript.

Kind regards,

Gennady S. Cymbalyuk, Ph.D.

Academic Editor

PLOS ONE

Journal Requirements:

Reviewers' comments:

Reviewer's Responses to Questions

**Comments to the Author**

1. Is the manuscript technically sound, and do the data support the conclusions?

Reviewer #1: Yes

Reviewer #2: No

2. Has the statistical analysis been performed appropriately and rigorously? 

Reviewer #1: Yes

Reviewer #2: No

3. Have the authors made all data underlying the findings in their manuscript fully available?

Reviewer #1: Yes

Reviewer #2: No

4. Is the manuscript presented in an intelligible fashion and written in standard English?

Reviewer #1: Yes

Reviewer #2: Yes

5. Review Comments to the Author

Reviewer #1: This paper describes the results of a pre-registered study on the impact of physical effort on decision deliberation. Authors report analyses of decision accuracy, response time, confidence, and metacognitive sensitivity (i.e. the similarity between decision accuracy and confidence). In contrast to authors prior assumptions, exerting effort while pondering decisions (here: standard perceptual decisions about the motion of moving dots) makes no difference for these variables, except response time (see below).

In brief, I have no critical concern with the study as it stands. Yes, reported results are negative results, but the authors’ conclusions are aligned with their data. Importantly, I see no major methodological flaw. Provided PLOS ONE accepts to publish negative results on pre-registered studies (which I think is important), then my conclusion is that this paper may be published almost as is.

Now, I think authors may improve the quality of their Introduction and Discussion sections. This is because it seems to me that they confuse two different types of effort (when exerted while performing some task): (i) effort that is instrumental to achieving the current goal, versus (ii) effort that is incidental to the task. For example, thinking hard to solve a reasoning problem or running fast to win a race are instrumental efforts that increase the likelihood of achieving the goal. In contrast, thinking hard while running a race or holding a hand grip while pondering a decision, are efforts that are incidental to the task. This distinction matters for understanding and/or predicting the effect of effort onto e.g., decision making (and decision-related variables). For example, one may expect that increasing instrumental effort may eventually increase decision accuracy and confidence (Lee and Daunizeau, 2021), whereas increasing incidental effort may decrease accuracy and confidence (e.g. because it may consume attentional or other resources that would be instrumental to the task). One may have different predictions, but I would still argue that one needs to distinguish instrumental and incidental types of effort. The issue here, is that authors’ reasoning seems to confuse these two types of effort, both when disclosing their hypotheses (Introduction) and when discussing their results. In my view, this renders their argument problematic at times. My advice is to systematically check and modify the Introduction and Discussion sections to disambiguate these two notions and their associated predictions/interpretations.

Jean Daunizeau.

Lee, D.G., and Daunizeau, J. (2021). Trading mental effort for confidence in the metacognitive control of value-based decision-making. ELife 10, e63282. https://doi.org/10.7554/eLife.63282.

Reviewer #2: In this study, Hagura et al., investigate the influence that making a physical effort during decision deliberation might have in decision accuracy and confidence. The interaction between decisions and actions is critical to fully understand decision making. However, I am not convinced that the experimental paradigm presented is enough to make any claim. Additional analysis and control are missing.

Main concerns:

- The study builds on the results of a pilot experiment in which metacognitive evaluation of perceptual decisions differed between force conditions. Supplementary Fig. 1 shows this data, but it seems that the test used to compare between the 3 force conditions was a t-test. When using a t-test to compare more than 2 groups Type I errors (ie. reject null hypothesis when it is true) accumulate. This could be the reason for finding significance in the metacognitive judgements in the pilot experiment but not in the main one. The appropriate tests to perform in this case are, for instance, an ANOVA or Kruskal-Wallis. A normality test should be performed to select the most appropriate one.

- It is not clear whether the strength needed to hold the joystick is effortful enough. In other words, there is no control on whether the motor cost associated with force and non-force conditions is significantly different. Only the mean and sd of the joystick’s deviation for each force condition is reported, but this does not ensure that the motor cost was of significance for subjects and this is critical to the experiment and the results. Although, this is stated in the discussion, without a control for that the results could be specific to the set-up, which would make them of low interest.

- Besides referencing a previous study, the ROC analysis of metacognitive reports should be explained in the text. Moreover, it might be useful to have the ROC analysis performed for each motion coherence separately. This might give some insight on whether there are differences, most likely, for low motion coherence conditions.

- Normality tests should be performed in all groups of data to select the most appropriate statistical test in each case.

- A link to the data is provided but it is not yet possible to access the data.

Minor comments:

- Age interval is missing.

- Some reference, for instance, Song et al., 2011 are referred but not listed.

6. PLOS authors have the option to publish the peer review history of their article (what does this mean?). If published, this will include your full peer review and any attached files.

Reviewer #1: **Yes: **Jean Daunizeau

Reviewer #2: No

---

## [Author Response · Author response to Decision Letter 0]

7 Sep 2022

Reviewer #1

Comment1: This paper describes the results of a pre-registered study on the impact of physical effort on decision deliberation. Authors report analyses of decision accuracy, response time, confidence, and metacognitive sensitivity (i.e. the similarity between decision

Accuracy and confidence). In contrast to authors prior assumptions, exerting effort while pondering decisions (here: standard perceptual decisions about the motion of moving dots) makes no difference for these variables, except response time (see below).

In brief, I have no critical concern with the study as it stands. Yes, reported results are negative results, but the authors’ conclusions are aligned with their data. Importantly, I see no major methodological flaw. Provided PLOS ONE accepts to publish negative results on pre-registered studies (which I think is important), then my conclusion is that this paper may be published almost as is.

Reply1: We appreciate the reviewer’s comment, acknowledging the value of reporting the negative result as an outcome of a pre-registered study.

Comment2: Now, I think authors may improve the quality of their Introduction and Discussion sections. This is because it seems to me that they confuse two different types of effort (when exerted while performing some task):

(i) effort that is instrumental to achieving the current goal, versus

(ii) effort that is incidental to the task. 

For example, thinking hard to solve a reasoning problem or running fast to win a race are instrumental efforts that increase the likelihood of achieving the goal. In contrast, thinking hard while running a race or holding a hand grip while pondering a decision, are efforts that are incidental to the task. This distinction matters for understanding and/or predicting the effect of effort onto e.g., decision making (and decision-related variables).

For example, one may expect that increasing instrumental effort may eventually increase decision accuracy and confidence (Lee and Daunizeau, 2021), whereas increasing incidental effort may decrease accuracy and confidence (e.g. because it may consume attentional or other resources that would be instrumental to the task). One may have different predictions, but I would still argue that one needs to distinguish instrumental and incidental types of effort. The issue here, is that authors’ reasoning seems to confuse these two types of effort, both

When disclosing their hypotheses (Introduction) and when discussing their results. In my view, this renders their argument problematic at times. My advice is to systematically check and modify the Introduction and Discussion sections to disambiguate these two notions and their associated predictions/interpretations.

Reply2: We appreciate the reviewer for pointing this out. We agree that the two different types of effort, instrumental and incidental, have not been well distinguished in the manuscript. In the revised manuscript, we have set this distinction clearer. 

We agree with the reviewer that, when the amount of effort is directly linked to the improvement of task performance (i.e. instrumental effort), it is straightforward to predict that higher effort expenditure will positively impact on the task performance and should lead to higher confidence. We now clarify that our key question relates to the unknown and interesting question; whether the task irrelevant, incidental effort can impact on the accuracy and the confidence of the decision. One example of such interactions is found in studies (also cited in our previously cited introduction) showing that (incidental) fluency can increase confidence. 

Another possibility is the one pointed out by the reviewer: incidental effort may distract the participant from the task. The attentional account also makes additional predictions for higher reaction time and lower accuracy under high incidental effort, which are clearly and empirically testable. Previous studies have shown that task irrelevant incidental effort indeed interacts with the ongoing task performance, not by just simply distracting the task. For example, Hagura et al., (2017) showed that the participants tend to avoid the option associated with higher (incidental) force exertion when performing a perceptual decision task, without any change in the sensitivity of their task performance. Similarly, Turner et al. (2020) demonstrated that the amount of exerted force level, which is announced to the participants after having made a particular perceptual decision, can paradoxically increase the confidence for the same decision decision. 

 Going one step further from these studies, our aim in the current study was to test how the incidental effort required to commit to the task can influence the task performance and the task confidence. Commitment is instrumental for initiating the task, but not instrumental for strength of perceptual evidence and thereby not for task performance. Indeed, faster reaction time observed during the force condition was not reflected on the accuracy measure, indicating that the effort used in our experiment was non-instrumental, thus incidental, to the primary task. 

 In the revised introduction (lines 58-70) and the discussion section (lines 342-345, 371-386) we have now included new paragraphs to unpack this issue. 

 

Reviewer #2: 

Comment 1: In this study, Hagura et al., investigate the influence that making a physical effort during decision deliberation might have in decision accuracy and confidence. The interaction between decisions and actions is critical to fully understand decision making. However, I am not convinced that the experimental paradigm presented is enough to make any claim. Additional analysis and control are missing.

Reply1: We appreciate the detailed and constructive comments of the reviewer.

Comment 2: The study builds on the results of a pilot experiment in which metacognitive evaluation of perceptual decisions differed between force conditions. Supplementary Fig. 1 shows this data, but it seems that the test used to compare between the 3 force conditions was a t-test. When using a t-test to compare more than 2 groups Type I errors (ie. Reject null hypothesis when it is true) accumulate. This could be the reason for finding significance in the metacognitive judgements in the pilot experiment but not in the main one. The appropriate tests to perform in this case are, for instance, an ANOVA or Kruskal-Wallis. A normality test should be performed to select the most appropriate one.

Reply2: We thank the reviewer for the critical evaluation of the pilot data statistics. We have now checked the normality of the data using Kolmogorov-Smirnov test. Data in all of the conditions did not violate the normality assumptions (p=0.138, p=0.2, p=0.2, for force 0N, 2.5N and 5N respectively). We also applied the requested ANOVA to the data (F(2, 13) = 2.21, p = 0.12). 

We agree with the reviewer that the analysis of the pilot experiment was done with a small sample size and relaxed a priori hypothesis about the direction of the effect. The purpose of the pilot experiment and data analysis was to explore the space of possible and likely hypotheses and narrow down this space with the discovery made by the pilot, which would eventually be formally tested in the main pre-registered experiment by setting the appropriate sample size for the predicted effect. Therefore, the analysis of the pilot experiment was exploratory in nature, aiming to maximise the sensitivity by relaxing the specificity. Given this aim, we believe that using paired t-test comparing the prominent two conditions in the pilot experiment was justified. 

For this reason, we believe that describing the result of the ANOVA in the manuscript will not benefit the paper but instead cause some confusion (Please note that we have made the full data set, including the pilot data, available for readers). In the revised manuscript, we have now further specified the aim of the pilot experiment in the Method section (lines 170-172). 

Comment 3: It is not clear whether the strength needed to hold the joystick is effortful enough. In other words, there is no control on whether the motor cost associated with force and non-force conditions is significantly different. Only the mean and sd of the joystick’s deviation for each force condition is reported, but this does not ensure that the motor cost was of significance for subjects and this is critical to the experiment and the results. Although, this is stated in the discussion, without a control for that the results could be specific to the set-up, which would make them of low interest.

Reply 3: We thank the reviewer for pointing this out, “whether the motor cost associated with force and non-force conditions is significantly different”. We should have explained this more clearly. 

First, the participants were required to maintain the position of the manipulandum at the home position to initiate the task, and during the task. In the force condition, the manipulandum was programmed to continuously produce 6N force in the direction away from the participant. Therefore, if the participants did not produce 6N toward the direction of one’s body to resist the manipulandum movement, the participants’ hand would be passively moved towards the direction of the force from the home position, or the manipulandum would slip away from the participants’ hand (and collide with the wall). So, without applying and maintaining 6N force, the trial would not start in the first place. 

In the no-force condition, the manipulandum did not produce any force. The participants did not have to resist any force. Therefore, in this setup, the amount of positional deviation of the manipulandum from the home position would directly indicate the failure to produce the required amount of force. The fact that the participants successfully maintained manipulandum position in the majority of trials (98.5%) indicates that the participants were indeed producing the force during the force condition. 

 Second, when the within-trial fluctuation of the manipulandum position during the task was compared between the force and no-force conditions, fluctuations were significantly more pronounced under the force condition than the no-force condition (p=0.019) (Figure S2). Since the variance of muscular activity is proportional to the size of its output (Signal Dependent Noise; Harris and Wolpert, 1997), this demonstrates that the participants were indeed producing more muscle activity (i.e. producing more effort) during the force condition. 

 Finally, in the force-condition, the reaction time generally decreased compared to the no-force condition, indicating that the force indeed gave impact to the participants’ behaviour, which made them to respond quickly to avoid the effort.

The evidence that (1) the manipulandum stayed at around the required position during the task even the force was applied, (2) within trial fluctuation of the manipulandum deviation was more prominent in the force condition, (3) participants tend to respond faster to avoid the force, clearly indicates that the motor cost was significantly higher in the force condition compared to the no-force condition. 

Now, we have clarified this point in the revised manuscript. Please see lines 271-286, together with the new Figures S2 presented in the Supplementary materials.

Comment 4: Besides referencing a previous study, the ROC analysis of metacognitive reports should be explained in the text. Moreover, it might be useful to have the ROC analysis performed for each motion coherence separately. This might give some insight on whether there are differences, most likely, for low motion coherence conditions.

Reply 4: Now, the detail of the ROC value calculation has been added to the manuscript. Please see line 241-250 of the text. 

Also, as the reviewer suggested, we have calculated the ROC analysis for different coherence levels. Note that ROC value calculation requires enough of both correct and incorrect trials, thus, for the high-coherence trials with extremely low probability of having incorrect trials, the value cannot be calculated. Therefore, instead of calculating for each coherence level, we divided the data into two groups, high coherence (12.8%, 25.6% 51.2%) and low coherence (3.2%, 6.4%) groups, and compared the ROC between conditions within each group. Consistent with the overall result, we found no difference between the two conditions (force vs. no-force) in each high and low coherence group (low; t(19)=0.69, p=0.50, high; t(19)=0.56, p=0.59). Now, this analysis has been added to the manuscript (lines 306-308) and in the Figure S3 of the Supplementary materials. 

Comment 5: Normality tests should be performed in all groups of data to select the most appropriate statistical test in each case.

Reply 5: We have now checked if the data fulfils the normality assumption for the parametric test. Please see lines 236-238.

Comment 6: A link to the data is provided but it is not yet possible to access the data.

Reply 6: The data is now made available. Please see <https://osf.io/be4u5/>. 

Comment 7: Age interval is missing. Some reference, for instance, Song et al., 2011 are referred but not listed.

Reply 7: Thanks for pointing them out. Now, they are fixed in the revised manuscript. See lines 159 and the reference section.

---

## [Decision Letter · Decision Letter 1]

21 Oct 2022

PONE-D-22-12208R1Does decision confidence reflect effort?PLOS ONE

Dear Dr. Hagura,

Thank you for submitting your manuscript to PLOS ONE. After careful consideration, we feel that it has merit but does not fully meet PLOS ONE’s publication criteria as it currently stands. Therefore, we invite you to submit a revised version of the manuscript that addresses the points raised during the review process.

Please, address the issues concerning validity of the statistical analysis raised by the reviewer.Please submit your revised manuscript by Dec 05 2022 11:59PM. If you will need more time than this to complete your revisions, please reply to this message or contact the journal office at plosone@plos.org. Please include the following items when submitting your revised manuscript:A rebuttal letter that responds to each point raised by the academic editor and reviewer(s). You should upload this letter as a separate file labeled 'Response to Reviewers'.A marked-up copy of your manuscript that highlights changes made to the original version. You should upload this as a separate file labeled 'Revised Manuscript with Track Changes'.An unmarked version of your revised paper without tracked changes. You should upload this as a separate file labeled 'Manuscript'.If applicable, we recommend that you deposit your laboratory protocols in protocols.io to enhance the reproducibility of your results. Protocols.io assigns your protocol its own identifier (DOI) so that it can be cited independently in the future. For instructions see: https://journals.plos.org/plosone/s/submission-guidelines#loc-laboratory-protocols. Additionally, PLOS ONE offers an option for publishing peer-reviewed Lab Protocol articles, which describe protocols hosted on protocols.io. Read more information on sharing protocols at https://plos.org/protocols?utm_medium=editorial-email&utm_source=authorletters&utm_campaign=protocols.

We look forward to receiving your revised manuscript.

Kind regards,

Gennady S. Cymbalyuk, Ph.D.

Academic Editor

PLOS ONE

Journal Requirements:

Reviewers' comments:

Reviewer's Responses to Questions

**Comments to the Author**

1. If the authors have adequately addressed your comments raised in a previous round of review and you feel that this manuscript is now acceptable for publication, you may indicate that here to bypass the “Comments to the Author” section, enter your conflict of interest statement in the “Confidential to Editor” section, and submit your "Accept" recommendation.

Reviewer #2: (No Response)

2. Is the manuscript technically sound, and do the data support the conclusions?

Reviewer #2: Partly

3. Has the statistical analysis been performed appropriately and rigorously? 

Reviewer #2: No

4. Have the authors made all data underlying the findings in their manuscript fully available?

Reviewer #2: Yes

5. Is the manuscript presented in an intelligible fashion and written in standard English?

Reviewer #2: Yes

6. Review Comments to the Author

Reviewer #2: I appreciate the effort made by the authors to improve the manuscript. I think it has gain in clarity. Still, few remaining comments:

Reply 2: I understand that the pilot experiment was only performed as an exploratory step before performing the experiment of the main paper. While this is completely fine and I am happy that it is now clearly stated in the text, I am still not convinced of using the t-test to compare the data. Under the experimental conditions that the authors are using, it is not correct to use such test. Thus, I would suggest to include the data as it is (in Fig S1), but to not include the result of a t-test. The authors could anyway discuss the tendency that they observed. Indeed, it should be clearly stated in lines 173-175 that there is a tendency.

Comment: Throughout manuscript, p values are reported by not the specific statistical test used. For instance, lines number 291, 295 or 297.

7. PLOS authors have the option to publish the peer review history of their article (what does this mean?). If published, this will include your full peer review and any attached files.

Reviewer #2: No

---

## [Author Response · Author response to Decision Letter 1]

27 Oct 2022

Response to the Reviewers

Reviewer #2

Comment1: I appreciate the effort made by the authors to improve the manuscript. I think it has gain in clarity. Still, few remaining comments:

Reply 2: I understand that the pilot experiment was only performed as an exploratory step before performing the experiment of the main paper. While this is completely fine and I am happy that it is now clearly stated in the text, I am still not convinced of using the t-test to compare the data. Under the experimental conditions that the authors are using, it is not correct to use such test. Thus, I would suggest to include the data as it is (in Fig S1), but to not include the result of a t-test. The authors could anyway discuss the tendency that they observed. Indeed, it should be clearly stated in lines 173-175 that there is a tendency.

Reply1: Following reviewer’s suggestions, we have revised the Supplementary Materials. First, we omitted the description of t-test from the legend of Figure S1, and also, deleted the asterisk (indicator of significance) from Figure S1b. 

Comment2: Throughout manuscript, p values are reported by not the specific statistical test used. For instance, lines number 291, 295 or 297.

Reply2: We have now added the name of the test together with the p-value, if not indicated. Full table of the statistical values is described in table S1.

---

## [Decision Letter · Decision Letter 2]

21 Nov 2022

Does decision confidence reflect effort?

PONE-D-22-12208R2

Dear Dr. Hagura,

We’re pleased to inform you that your manuscript has been judged scientifically suitable for publication and will be formally accepted for publication once it meets all outstanding technical requirements.

Kind regards,

Gennady S. Cymbalyuk, Ph.D.

Academic Editor

PLOS ONE

Additional Editor Comments (optional):

Reviewers' comments:

Reviewer's Responses to Questions

**Comments to the Author**

1. If the authors have adequately addressed your comments raised in a previous round of review and you feel that this manuscript is now acceptable for publication, you may indicate that here to bypass the “Comments to the Author” section, enter your conflict of interest statement in the “Confidential to Editor” section, and submit your "Accept" recommendation.

Reviewer #2: All comments have been addressed

2. Is the manuscript technically sound, and do the data support the conclusions?

Reviewer #2: Yes

3. Has the statistical analysis been performed appropriately and rigorously? 

Reviewer #2: Yes

4. Have the authors made all data underlying the findings in their manuscript fully available?

Reviewer #2: Yes

5. Is the manuscript presented in an intelligible fashion and written in standard English?

Reviewer #2: Yes

6. Review Comments to the Author

Reviewer #2: The manuscript has significantly improved and all my comments have been addressed. I have no further concerns.

7. PLOS authors have the option to publish the peer review history of their article (what does this mean?). If published, this will include your full peer review and any attached files.

Reviewer #2: No

---

## [Editor Report · Acceptance letter]

28 Nov 2022

PONE-D-22-12208R2 

Does decision confidence reflect effort? 

Dear Dr. Hagura:

I'm pleased to inform you that your manuscript has been deemed suitable for publication in PLOS ONE. Congratulations! Your manuscript is now with our production department. 

Kind regards, 

on behalf of

Dr. Gennady S. Cymbalyuk 

Academic Editor

PLOS ONE